# Fabric tearing performance state perception and classification driven by multi-source data

**Jianmin Huang[1,2], Qingchun Jiao [iD][3]\*, Yifan Zhang[3], Gaoqing Xu[1,2], Lijun Wang[3], Dong Yue[4]**

1 Zhejiang Institute of Standardization (Zhijiang Standardization Think Tank), Hangzhou, China,
2 Technology Innovation Center of State Market Regulation Management (Research and Application of Digital Market Regulation), Hangzhou, China, 3 School of Automation and Electrical Engineering, Zhejiang University of Science and Technology, Hangzhou, China, 4 Zhejiang Light Industrial Products Inspection and Research Institute, Hangzhou, China

\* jiaoqch@zust.edu.cn

## Abstract

The tear strength of textiles is a crucial characteristic of product quality. However, during the laboratory testing of this indicator, factors such as equipment operation, human intervention, and test environment can significantly influence the results. Currently, there is a lack of traceable records for the influencing factors during the testing process, and effective classification of testing activities is not achieved. Therefore, this study proposes a state-awareness and classification approach for fabric tear performance testing based on multi-source data. A systematic design is employed for fabric tear performance testing activities, which can real-time monitor electrical parameters, operational environment, and operator behavior. The data are collected, preprocessed, and a Decision Tree Support Vector Machine (DTSVM) is utilized for classifying various working states, and introducing ten-fold cross-validation to enhance the performance of the classifier, forming a comprehensive awareness of the testing activities. Experimental results demonstrate that the system effectively perceives fabric tear performance testing processes, exhibiting high accuracy in the classification of different fabric testing states, surpassing 98.73%. The widespread application of this system contributes to continuous improvement in the workflow and traceability of fabric tear performance testing processes.

## Introduction

Textiles are widely used in daily life and industrial fields, and their strength is one of the important indicators to evaluate their quality and reliability [1]. Tear performance can evaluate the endurance and strength of fabrics under tearing stress [2, 3], providing a reference for the design and process improvement of textile products. Therefore, the standardization of tear performance testing experiments is particularly important. When fabric performance testing laboratories carry out such activities, they are easily affected by factors such as equipment operation, personnel operation, and test environment. It is necessary to establish traceability records of various influencing factors during the experimental process. However, currently,

**Data Availability Statement:** In order to comply with PLOS ONE's policy on code sharing, we have uploaded all the necessary code, datasets and related materials generated in the course of this research work to the GitHub platform. These

resources are publicly available at the following link: https://github.com/yuyuyu123YUYUYU/data.

**Funding:** Science and Technology Planned Project of the State Administration for Market Regulation (No.CY2023213);"Chu Ying" Project (Core Project) of Zhejiang Administration for Market Supervision (No.2022MK057);Natural Science Foundation of Zhejiang Province (No.LGG20F020008). The funders had no role in study design, data collection and analysis, decision to publish, or preparation of the manuscript.

**Competing interests:** The authors have declared that no competing interests exist.

there is no comprehensive system that can perceive and manage testing activities comprehensively, accurately, and in real time.

This paper draws on the system design of the situation awareness system and proposes a research on state awareness and classification of fabric tear performance detection driven by multi-source data. The concept of "situational awareness" first originated from the U.S. Air Force, which refers to the awareness and understanding of various elements or objects in the environment and the prediction of future states at a specific time and space [4, 5]. The situation awareness process can be roughly divided into three main processes: awareness, understanding, and prediction. This article starts from the development needs of fabric performance testing laboratories for effective monitoring of testing activities, introduces situational awareness technology to establish a system for sensing the situation of fabric tear performance testing activities, and analyzes a large number of real-time and dynamic testing equipment operation data and testing laboratory basic data. Fusion analysis [6, 7] extracts effective information to achieve comprehensive control over the operating status of detection equipment. Due to the different problems solved, this article only draws on the two process ideas of awareness and understanding in situational awareness, traces the working status and data of the equipment during the detection process, and realizes real-time awareness of the situation of fabric tear performance detection activities.

This system can collect information from a variety of different data sources and comprehensively analyze it to form situational awareness of the overall detection activities. The research content includes the following aspects:

(1) For fabric tearing performance testing activities, build a non-invasive real-time situational awareness system for equipment power parameters, environmental temperature and humidity, and operator behavior to achieve full control of the equipment's operating status without affecting the normal experimental process.

(2) Use the decision tree support vector machine method to classify the preprocessed multi-source data into multiple working states to achieve traceability records of fabric tearing performance testing activities.

In summary, the state-awareness system for fabric tear performance testing driven by multi-source data has significant application prospects in the textile industry.

The system accurately identifies the dynamic situation of fabric tear performance testing activities and possesses good real-time performance and scalability. The rest of the paper is structured as follows." The "Related work" section discusses related work, outlines the fabric strength testing methodology, and summarizes the multi-source data-driven situational awareness technique." The "Research methodology" section describes the research methodology in detail and presents the system components of the situational awareness system developed in this paper as well as the sensing and understanding aspects." The "Experimental Design" section outlines the experimental design, describes the field setup of the system and the multi-source data-driven situational awareness experiments on different fabric materials." The "Experimental Results and Discussion" section analyzes the results of the experiments by comparing different fabric materials. The "Conclusion and Outlook" section summarizes the experimental results and provides an outlook on the future development of the system.

## Related work

### Fabric strength Testing methods

Fabrics often break due to local damage caused by concentrated loads during use [8]. Fabric strength is one of the important indicators for evaluating the durability of fabrics [9, 10], including tear strength, breaking strength, rupture strength and peel strength. Taking the tear strength test as an example, the three principles for fabric tear research are: the starting point of tearing is located at the crack, that is, the stress concentration point [11]; the external force should be an impact load (force with acceleration) [12]; During the tearing process, the sheared yarn is perpendicular to the clamped yarn, and the tearing force is transmitted through the interweaving resistance of the fabric. Only when the interweaving resistance is less than the tearing force can the tearing proceed [13]. Liang Yonghong et al. [14] used Ningbo Fangyi YG026H multi-functional electronic fabric strength machine to conduct applicability analysis on the method of testing the tear strength of woven textiles. Guan Xiaoyu et al. [15] used a double-column floor-standing electronic universal testing machine. UTM4000 analyzes and studies the tearing mechanical properties. This article is based on the international standard "ISO 13937-2-2000 Textile fabrics—Tear properties—Part 2: Determination of tear strength of trouser specimens (single seam)", using the CRE isokinetic elongation tester to conduct trousers shape testing To test the tearing strength of sample fabrics, for each laboratory fabric sample, 2 sets of samples should be cut, one set is radial and the other is weft direction. The samples are rectangular, with a length of (200±2) mm and a width of (50±1)mm. Each sample should be cut with a slit of (100±1) mm parallel to the length direction from the center of the width direction, and the tear end point should be marked at a distance of (25±1) mm from the uncut end of the sample, such as As shown in Fig 1. The principle of testing the fabric tear strength of a trouser-shaped sample is to clamp the two legs of the trouser-shaped sample so that the sample incision line is in a straight line between the upper and lower clamps. Set the gap length of the fabric tear tester to 100mm, and pull The elongation rate is 100mm/min. Start the instrument and apply tensile force in the direction of the incision, as shown in Fig 2. Record the tearing strength until the tear reaches the specified length, and calculate the tearing strength based on the peak value on the curve drawn by the automatic drawing device. During the sample preparation and experiment, the operator should always pay attention to the preparation and tearing of the sample. By testing and analyzing the tearing strength of the sample, the quality of the fabric can be effectively evaluated [16], providing an important basis for production and quality control.

Guifei Wu et al. [17] conducted a study on the influence of temperature and humidity on the physical and mechanical properties of fabrics in the context of fabric performance testing activities. They found that changes in temperature and humidity can affect the tear strength performance of fabrics. Therefore, it is crucial to pay special attention to the impact of temperature and humidity variations during the experimental process. The operational status of equipment can be reflected through electrical parameters. Fabricio M. A et al. [18], in their work, analyzed time-series electrical parameters to detect the operating status of monitored machines. This approach achieved independent, parallel, and non-intrusive monitoring. Prior to this, in fabric performance testing activities, no one had established a non-intrusive situational awareness monitoring system capable of monitoring influencing factors.

### Multi-source data-driven situational awareness technology

Multi-sensor multi-source data fusion [19, 20] is a process of utilizing computer technology to automatically analyze and integrate information and data from multiple sensors according to

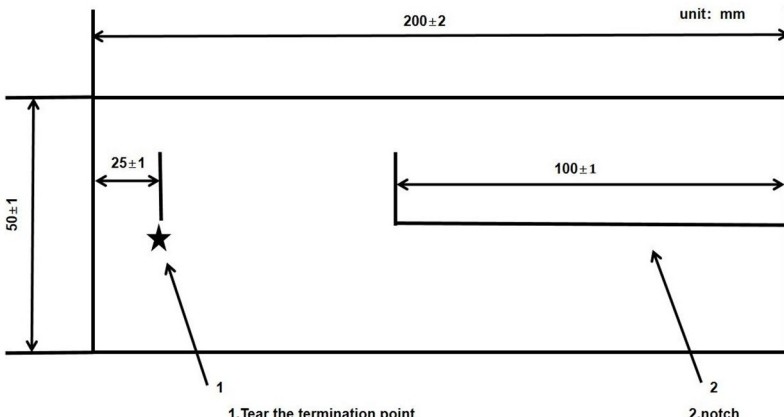

**Fig 1. Trouser-shaped fabric sample.** The samples are rectangular, with a length of (200±2) mm and a width of (50 ±1)mm. Each sample should be cut with a slit of (100±1) mm parallel to the length direction from the center of the width direction, and the tear end point should be marked at a distance of (25±1) mm from the uncut end of the sample.

certain rules. This optimization process is carried out to achieve the required decision-making and estimation. Similar to human perception, different sensors have irreplaceable roles, and various sensors perform multi-level, multi-space data information complementation, and optimized combination processing, ultimately generating a consistent interpretation of the monitored environment. For situational awareness systems, sensor products serve as integral components of the core system for data collection. The fusion of multi-sensor data information achieves diversification in information collection, complementary and optimized combination of multi-source data. This enhances the detection accuracy and precision of the perception layer module in situational awareness systems, providing crucial data support for situational awareness systems [21].

In terms of the abstraction level of information processing, multi-sensor fusion in structure can be broadly categorized into three levels: (1) Data-level fusion: Various input data from sensors are fused, and feature vectors are extracted from the fused data for judgment and recognition. (2) Feature-level fusion: This belongs to an intermediate fusion level, involving the extraction of input data from sensors, followed by comprehensive analysis and processing of feature information. (3) Decision-level fusion [22]: This is a high-level fusion where each sensor combines the initial results generated from various data sources to generate a fused result and make a judgment. Decision-level fusion can be further classified into serial decision-level fusion and parallel decision-level fusion: Serial decision-level fusion involves the aggregation of decision results from various decision-makers in a predetermined sequence and according to certain rules. Each stage of decision-making relies on the output of the previous stage. This fusion method can control the decision-making process but may suffer from issues such as information lag and local optima. Parallel decision-level fusion, on the other hand, refers to independent decision-making by each decision-maker simultaneously. Their individual decision results are then compared and synthesized to obtain a comprehensive decision result. This approach can leverage the expertise and experience of each decision-maker but may lead to increased complexity in the decision-making process and challenges in achieving decision result consistency.

Based on the theory of multi-sensor fusion, selecting an appropriate classifier for multi-classification problems is a promising solution. In recent years, various improved Support Vector

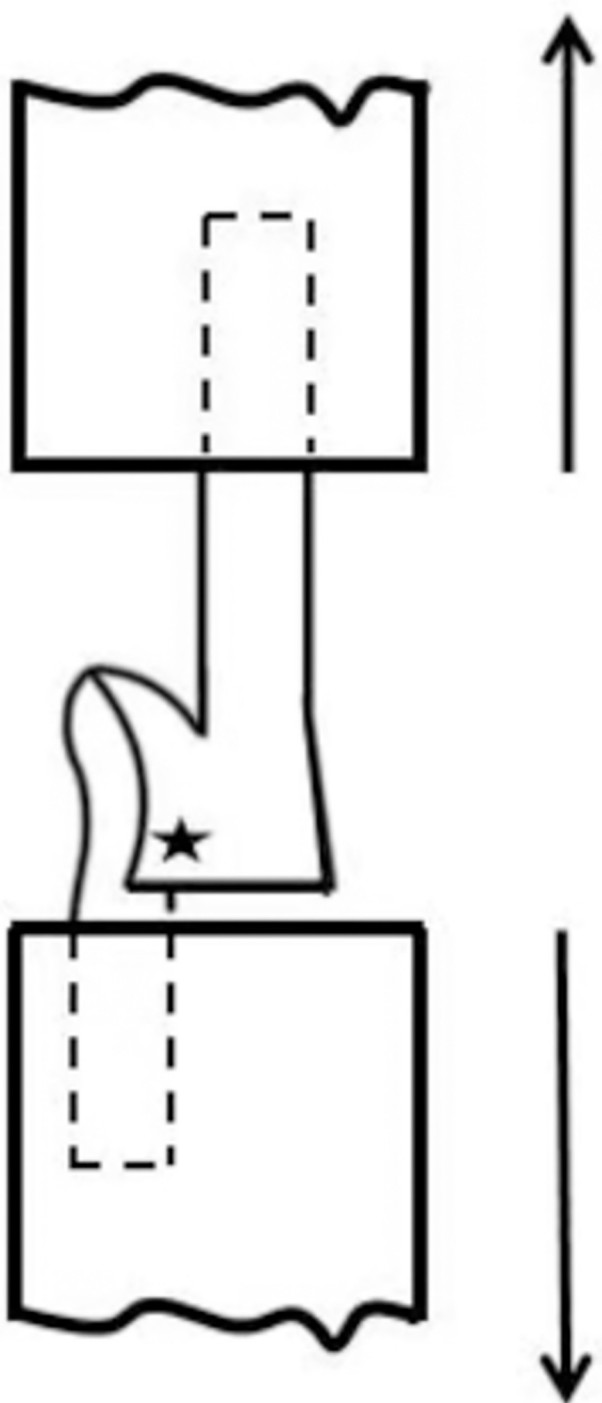

**Fig 2. Tear strength testing schematic.** The principle of fabric tearing strength detection for trouser sample is to clamp the two legs of trouser sample so that the cut line of the sample is in a straight line between the upper and lower fixture, and the pulling force is applied to the cut direction by starting the instrument.

Machines (SVMs) have shown excellent performance compared to other classification algorithms. However, most of them are binary classification models and cannot be directly applied to multi-classification problems. A binary decomposition strategy can solve a multi-classification problem, i.e., decompose the original multi-class classification problem into a set of binary subproblems of lower complexity. For each subproblem, corresponding binary classifiers can be easily derived using an appropriate learning algorithm. The outputs of all these classifiers are then combined to classify the labels of new objects. This divide-and-conquer strategy allows for simpler learning of complex decision boundaries for multi-class classification problems than directly building a unique classifier that separates all classes simultaneously. One-against-all (OAA) and one-against-on (OAO) are two widely adopted binary decomposition methods. For a K-class problem, the OAA decomposition strategy splits each class and all other classes into two groups and subsequently introduces K parallel binary classifiers. The OAO decomposition builds a binary classifier B between each pair of classes (i, j) and generates K(K-1)/2 parallel binary classifiers. In addition to this, Li J [23] proposed a multi-classification algorithm based on a dual support vector machine decision tree to address security and privacy issues in IoT data. Gai R et al. [24] presented a river water quality assessment method using an improved grey relational analysis (ACGRA) and particle swarm optimization multi-class support vector machine (PSO-MSVM). This method is applied to evaluate the environmental quality of river water. Chen Z et al. [25] addressed the fault diagnosis problem of power transformers by proposing a reverse merging strategy. They improved the SVM model of the classification decision tree to enhance the training efficiency and classification accuracy of the diagnostic model. From these studies, it can be concluded that various SVM approaches are widely applied to multi-classification problems in different domains, showing significant potential. At the same time, scholars have also done detailed research on SVM optimization to get the best classification results. Kalita D J et al. [26] proposed a new framework for constructing intrusion detection systems (IDS) in non-stationary environments, and trained support vector machines (SVMS) for intrusion detection by dynamically optimizing their hyperparameters and dynamically. Aiming at the problem that early machine fault diagnosis will be affected by environmental noise, Zhao W et al. [27] proposed a data-driven early fault diagnosis method based on vibration signal based on reinforcement learning (RL) optimization support vector machine (SVM) model, and explored the use of reinforcement learning hyperparameter optimization to improve the performance of SVM. Ding S et al. [28] proposed a fault diagnosis method for motor drive systems based on multi-class support vector machine (SVM) classifiers, hybrid particle swarm optimization (PSO) and gravity search algorithm (GSA). This method combines the global search capability of PSO and the local search capability of GSA. Combining the advantages of PSO and GSA, the hybrid GSAPSO algorithm is used to optimize the multi-class SVM classifier, which improves the classification performance. Considering the multi-classification problem in the fabric tear performance testing activity addressed in this paper, it is evident that SVMs are a promising solution with substantial application prospects.

In summary, existing relevant works have laid the theoretical and practical foundations for a situational awareness system in fabric tear performance testing driven by multi-source data. This system can be applied effectively in the textile industry for fabric testing.

## Research methodology

Faced with challenges in the experimental process of fabric tear performance testing, such as issues related to detection states and data traceability, and recognizing the lack of non-intrusive system perception, this paper stems from the developmental needs of textile strength

testing in the fabric testing laboratory. Starting from the needs of fabric testing laboratories for testing quality control, this paper introduces situational awareness technology into the field of textile strength testing and designs a situational awareness system for fabric tearing performance testing. The research methods in this section include system composition, situational perception and situational understanding.

The 'System Composition' section introduces the construction process of the situational awareness system and the establishment of the database. The 'Situational Perception' section introduces data collection and data preprocessing processes. The 'Situational Understanding' section introduces the Decision Tree Support Vector Machine (DTSVM) algorithm model developed for fabric tear performance testing. It combines the hierarchical structure of a decision tree with the classification capabilities of a support vector machine. The model collects and processes real-time dynamic data from the situational awareness system, enabling comprehensive monitoring of the operational status of the detection equipment. Furthermore, it effectively addresses the issue of data traceability in fabric tear performance testing.

## System composition

The situational awareness system designed in this paper for fabric tear performance detection, driven by multi-source data (referred to as SASFTPD-MSD), is illustrated in Fig 3. SASFTPD-MSD consists of two levels: situational perception and situational understanding. In the situational perception part, multiple sensors placed around the fabric tear testing machine collect data from various sources, generating raw multidimensional data. Subsequently, after processes such as data cleansing, sliding window sampling, and data normalization, multidimensional sequential data is obtained. The situational understanding part classifies the operational

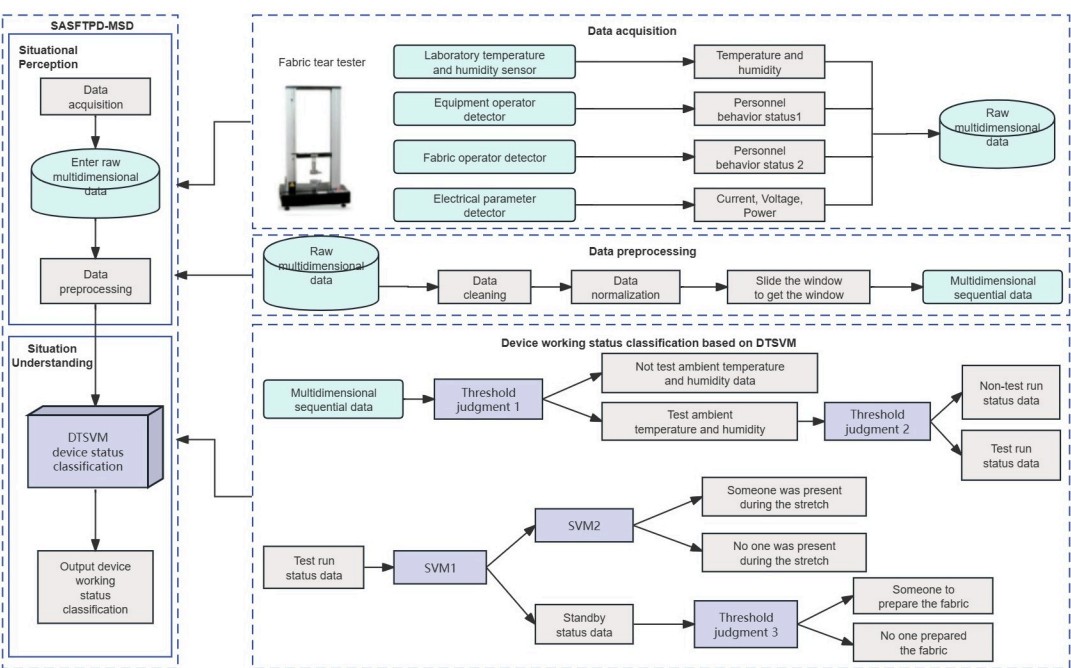

**Fig 3. SASFTPD-MSD system architecture.** SASFTPD-MSD consists of two levels: Situational Perception and Situational Understanding. In the 'Situational Perception' part preprocesses the multi-source data of the multi-source sensors arranged around the fabric tear tester, and the 'Situational Understanding' part classifies the working state of the multi-dimensional sequence data based on the DTSVM model.

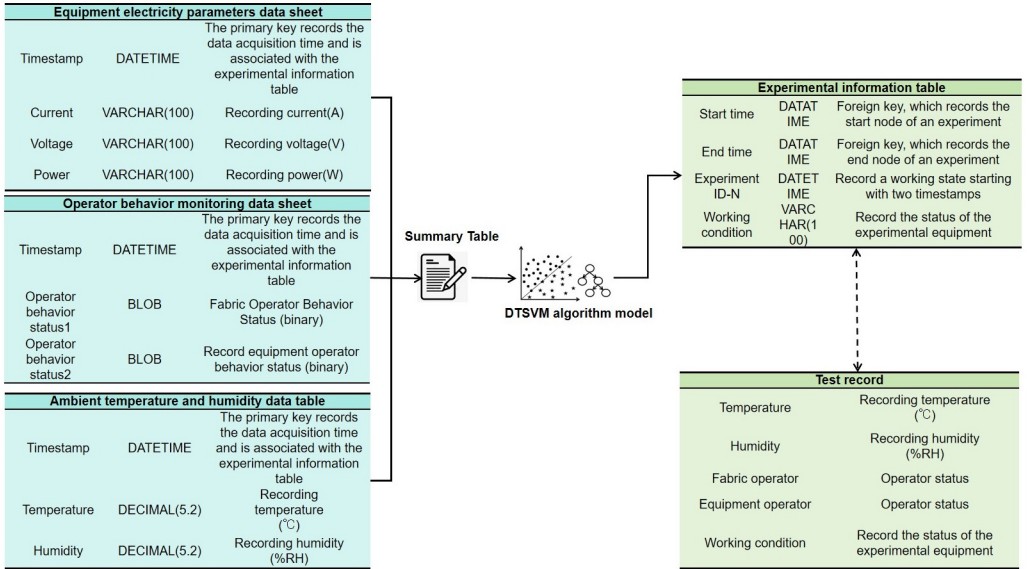

**Fig 4. Data flow.** The data tables for equipment power parameters, operator behavior monitoring, and environmental temperature and humidity are aggregated into a master table through time alignment. Subsequently, the data from the master table undergoes preprocessing before being fed into the DTSVM algorithm model. The result is the generation of an experimental information table, providing the working states of the experimental equipment within a specific experiment's start and end time nodes. This setup enables a bidirectional data flow display with the detection record table.

states of the equipment based on the DTSVM model applied to the multidimensional sequential data, achieving overall situational awareness for fabric tear performance detection.

The data flow is illustrated in Fig 4, where the data tables for equipment power parameters, operator behavior monitoring, and environmental temperature and humidity are aggregated into a master table through time alignment. Subsequently, the data from the master table undergoes preprocessing before being fed into the DTSVM algorithm model. The result is the generation of an experimental information table, providing the working states of the experimental equipment within a specific experiment's start and end time nodes. This setup enables a bidirectional data flow display with the detection record table.

## Situational perception

### Data collection.

1. *Power parameter monitoring*. The fabric tear testing machine conducts trouser-shaped specimen tear tests using the CRE constant-rate elongation tester. For fabric tear performance testing, real-time monitoring of the power usage of the fabric tear testing machine is conducted. This includes parameters such as current, voltage, and power. Throughout the fabric tear performance testing process, the operational states of the CRE constant-rate elongation tester are categorized as standby, stretching, resetting, and sample changing, as detailed in Table 1. Considering these states, the collection of power parameters needs to span the entire experimental process. Particularly during the stretching operation of the equipment, the current and power consumption in the power parameters are maximal. Furthermore, according to international standards 'ISO 13937-2-2000 Textile fabrics—Tear properties—Part 2: Determination of tear strength of trouser specimens (single seam)', for constant-rate elongation instruments, if force and elongation records are obtained through

**Table 1. Operational states of the CRE constant-rate elongation tester.**

| Working Status | Description |
|---|---|
| Standby mode | The equipment is started and ready for experiments, waiting for operating instructions |
| Stretched state | Measure the tensile properties of fabrics, including tensile strength, elongation, etc. |
| Reset state | After the experiment is completed, return to the initial position and prepare for the next round of experiments |
| Sample changing state | Replace experimental samples and install new samples |

data acquisition chips and software, the data acquisition frequency should be at least 8Hz. Following the Nyquist sampling theorem, to faithfully reconstruct the original signal without distortion, the sampling frequency should be greater than twice the highest frequency of the signal. Therefore, the designed frequency for collecting power parameters should be no less than 16Hz to real-time reflect changes in monitoring equipment power loads.

2. *Temperature and humidity monitoring.* Environmental temperature and humidity can impact the physical properties of fabrics. In high-temperature environments, the strength and toughness of fabrics decrease, while in humid conditions, the softness and elasticity of fabrics decrease. Fabric tear performance testing requires testing under specific temperature and humidity conditions. The international standard 'ISO 139-1973 Textiles—Standard atmospheres for conditioning and testing' stipulates the atmospheric conditions for testing: an atmospheric temperature of 20.0℃ with a tolerance of ±2.0%, and a relative humidity of 65.0% with a tolerance of ±4.0%. To obtain more accurate test results, it is necessary to install temperature and humidity sensors in the testing area of the fabric tear performance testing equipment. This helps prevent the test results from being affected by local temperature and humidity conditions. Therefore, when selecting the model of temperature and humidity sensors, sensors with an accuracy of 0.1 should be chosen.

3. *Operator behavior detection.* In fabric tearing performance testing experiments, it is necessary to monitor in real-time whether individuals are present during different operational stages. Based on the four working states of individuals during equipment operation, as shown in Table 2, which are: sample preparation stage, sample humidity adjustment stage, sample experimentation stage, and data recording stage, two types of personnel presence need to be monitored. One is the monitoring of the fabric operator's behavior state, and the other is the monitoring of the equipment operator's behavior state and data recording. The fabric operator needs to be present throughout the sample preparation and data recording, while the equipment operator needs to be present throughout the sample experimentation.

**Table 2. Four working states of personnel during equipment operation.**

| Working Status | Description |
|---|---|
| Sample preparation stage | Prepare samples, one set of radial direction and one set of weft direction, 5 samples in each group |
| Sample humidity conditioning stage | The sample is placed in a standard atmospheric environment for humidity control. When the weight gradient of the sample does not exceed 0.25%, it is considered to have reached an equilibrium state. Generally, the humidity control time of textiles exceeds 24 hours |
| Sample experiment stage | Including the stretching state, reset state and sample changing state of the equipment |
| Data recording stage | Record data collected during testing and generate test reports |

**Data preprocessing.** In response to the multi-source sensor data collected during the data acquisition stage, three major categories are obtained: power parameter monitoring data, environmental temperature and humidity monitoring data, and operator behavior detection data. Due to the different sampling frequencies of various sensors and the prolonged duration of fabric tear performance testing activities, the collected data are all time-series data. Therefore, preprocessing operations such as data cleaning, data normalization, and sliding window segmentation are required for the collected data.

1. *Data cleaning.* Data cleaning involves noise reduction and data alignment.
   Noise Reduction:Since multi-source sensor data are collected from different sensors, the data collected from laboratory temperature and humidity sensors and operator detection devices may be subject to interference from the environment or their own performance, leading to data disturbances. For instance, long working hours of temperature and humidity sensors can cause data drift. To address this issue, median filtering is employed to eliminate pulse signal anomalies in the collected data. Median filtering, a non-linear signal processing method, utilizes a moving window with an odd number of points. The central value in the window is replaced with the median of the values within the window. By determining whether the signal value is a local maximum or minimum within the filtering window, median filtering effectively removes pulse noise and abrupt changes, as shown in Eq (1).

$$\hat{x}i = \text{median} \, W_i = \text{median}(x_{i-N+1}, x_{i-N+2}, \ldots, x_i, \ldots, x_{i+N-1}), \tag{1}$$

Here, $\hat{x}_i$ represents the denoised data, $N$ is the window size, and $i$ is the starting position of the window. Operator detection devices are typically used to determine the presence of individuals in a specific area. Their output is usually a Boolean value (0 or 1), indicating the absence or presence of a fanyperson. However, these devices may generate frequent false alarms or sustained erroneous states. Although the output of operator detection devices is in Boolean form, continuous detection results can be smoothed using logical AND operations in signal smoothing techniques. This helps reduce false positives and false negatives. If the detection result is consistently '1' for k consecutive times, it is confirmed as a presence of an individual. This can be represented by Eq (2).

$$\mathbf{y_t} = \begin{cases} 1, & \text{if } x_t = x_{t-1} = \cdots = x_{t-k+1} = 1 \\ 0, & \text{otherwise} \end{cases}, \tag{2}$$

Where $x_t$ is the original detection result (Boolean) for theinstance, and $y_t$ is the result after smoothing. Data Alignment:Data alignment involves interpolating the data collected from different sensors based on the start and end times of the time window, ensuring that the data from different sensors is not affected by the timing of the collection. Additionally, due to prolonged sensor operation leading to slower response times, missing values may occur during data collection. Linear interpolation is employed to fill in these missing values, as shown in Eq (3).

$$x[n] = x[a] + \frac{x[b] - x[a]}{b - a} * (n - a) \tag{3}$$

Here, represents missing points, and the window size is $N = [a, b]$. Once the data from all sensors has been interpolated to the same time interval, they can be considered synchronized.

2. *Data normalization*. The laboratory environmental temperature and humidity monitoring data are based on the threshold set for experimental temperature and humidity under the standard atmosphere mentioned in international standards to distinguish between normal and abnormal conditions during the experiment. The data from the operator behavior detection is boolean data, which is also judged based on the set threshold. Since different electrical parameters have different dimensions (voltage is measured in volts, current in amperes, and power in watts), their diverse dimensions can impact data analysis and processing. Normalization is employed to transform them into a uniform dimensionless indicator range, facilitating data analysis and comparison. This paper uses the most commonly used method in data processing, namely Min-Max normalization. The original data is linearly transformed using Eq (4), mapping the results to the range [0, 1].

$$x_i' = \frac{x_i - \min(W_i)}{\max(W_i) - \min(W_i)} \tag{4}$$

3. *Sliding window processing*. In the overall activity design for fabric tearing performance testing, the sliding window technique is used to process and analyze the real-time collected time-series data. Let the minimum sampling frequency be $f_{min}$ Hz, which corresponds to a sampling period $T_{sample} = f_{min}$ second. Accordingly, the size of the sliding window N can be set to one or more complete sampling periods, which is expressed by Eq (5).

$$\mathbf{N} = \mathbf{k} \cdot T_{sample} = k \cdot \frac{1}{f_{min}} \tag{5}$$

where *k* denotes the number of complete sampling cycles selected, determined according to the actual demand.

To ensure continuity and capture subtle changes in the data, we set the step size S of the sliding window to half the size of the window, which is expressed by Eq (6).

$$S = \frac{N}{2} = \frac{k}{2} \cdot \frac{1}{f_{min}} \tag{6}$$

When applying this sliding window algorithm, for any time point *t*, the current sliding window $w_t$ as well as the position of the next window can be defined on the data series. The range of the *i* th window is $[t_i - S, t_i]$, and the starting position of the next window $w_{i+1}$ is updated to $t_i + S$.

The advantage of the sliding window approach is that, for a large number of continuous real-time data sets, it is able to perform segmentation while maintaining data continuity, which facilitates the extraction of potential pattern and trend information. This means that the system can update the data sub-sequence within the sliding window in real-time, and monitor and analyze it, so as to effectively identify and track the changes in the working status of the fabric tearing performance testing activities, and enhance the accuracy and traceability of the testing process.

**Situation understanding.** Based on the problem of classifying equipment operation states in fabric tear performance testing addressed in this paper, and given that the input involves multi-source data, the classification structure is not a simple binary classification problem. Therefore, a DTSVM multi-class method has been proposed, representing the optimal solution for classifying equipment operation states within the situational awareness system. It is a parallel decision-level fusion method where multiple sensor data undergo parallel decision-level

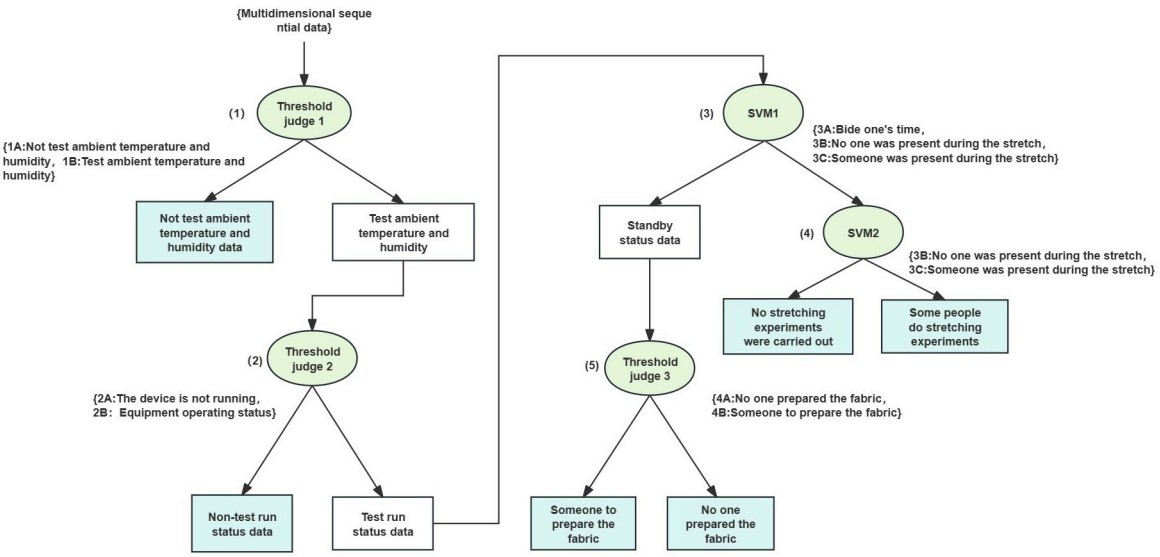

**Fig 5. DTSVM algorithm model structure.** It is a parallel decision-level fusion method where multiple sensor data undergo parallel decision-level fusion. The method employs multiple threshold detectors and support vector machine models, executing sequential decisions. Each decision stage relies on the output of the preceding stage and operates in the form of a decision tree. Each model receives distinct input data, generates corresponding classification results, and the final decision results are fused through the outputs of multiple model.

fusion. The method employs multiple threshold detectors and support vector machine models, executing sequential decisions. Each decision stage relies on the output of the preceding stage and operates in the form of a decision tree. Each model receives distinct input data, generates corresponding classification results, and the final decision results are fused through the outputs of multiple models.

As illustrated in Fig 5 of the DTSVM algorithm structure, the process of the DTSVM algorithm model structure is as follows: (1)Input the collected 7-dimensional vector dataset into the first threshold discriminator to classify non-1A test environment temperature and humidity vector data and 1B test environment temperature and humidity vector data.(2)Input the 1A test environment temperature and humidity vector data into the second threshold discriminator to classify 2A non-test operating state vector data and 2B test operating state vector data.(3) Input the 2B test operating state vector data into the first SVM to classify 3A standby state vector data and non-standby state vector data.(4)Input non-standby state vector data into the second SVM to classify 3B unmanned stretching experiment state and 3C manned stretching experiment state.(5)Input the standby state vector data into the third decision tree to classify 4B manned fabric preparation state and 4A unmanned fabric preparation state.(6)Construct the DTSVM algorithm model based on the above content. Through analysis, the working states can be categorized into 6 classes.

**Decision tree support vector machine.** There is a difference in average similarity between samples of different classes, and for a classification problem with m classes, the DTSVM is a binary tree with m-1 internal nodes, each of which is an m leaf node and a binary classifier. If the classification performance of the upper nodes of the decision tree is not good, the overall classification performance becomes worse. Therefore, to ensure high generalization ability, the most separable classes should be separated preferentially at the upper nodes of the decision tree. In DTSVM, if each internal node considers using both binary SVM classifiers and

thresholding judgments, the process of constructing the model can be regarded as a hybrid strategy, and the selection of each node may be based on the optimal classification hyperplane of SVM or on the threshold division of a certain feature.

For each internal node of the decision tree, suppose that a division based on the threshold t of feature A is considered, when the optimal feature and threshold can be selected based on some measure (e.g., information gain, Gini index, etc.), which can be used to reduce the uncertainty of the node. If the division is based on the threshold value using Eq (7):

$$\text{Optimal  Threshold} = \arg\max_{t} \text{Gain}(A, t) \tag{7}$$

where *Gain*(*A*, *t*) denotes the gain from dividing the dataset by feature *A* at threshold *t*.

Suppose we consider training a binary SVM classifier to classify the two classes of samples, denoted by Eq (8):

$$\underset{w,b}{\text{minimize}} \quad \frac{1}{2}||w||^2 \quad \text{subject  to} \quad y_i(w^T x_i + b) \geq 1, \quad \forall i \in S \tag{8}$$

where *w* is the normal vector of the hyperplane, *b* is the displacement term, $x_i$ is the feature vector, $y_i \in \{-1, 1\}$ is the category label, and *s* is the set of training samples contained in the current internal node.

The specific construction process of the DTSVM model takes place iteratively, and each node is divided based on a specific optimization criterion to decide whether to use the SVM or the threshold judge.

The model was chosen for multi-source data and multiple problems because of its ability to integrate the intuitiveness and interpretability of decision trees and the generalization ability and nonlinear classification advantages of SVM. In complex systems, their state classification is often not a simple binary distinction, but involves multiple categories, and there may be complex boundaries and intrinsic connections between the categories. Individual nodes can either be filtered for classification using a threshold judge, or SVM can be applied for classification. Multiple models are connected in series through the form of decision tree, and each model receives part of the input data and outputs the classification results, and finally the outputs of all models are integrated to realize the effect of integrated learning, which improves the overall classification performance and robustness.

**Decision tree method.** The Decision Tree method is a commonly used machine learning algorithm employed for classification and regression tasks. This model systematically partitions data features, ultimately generating a tree-shaped structure that can be utilized for decision-making. The fundamental idea behind a decision tree is to start from the root node and, through a series of branching nodes, classify or predict data. Each branching node corresponds to a feature, and the different paths of the branches represent different values of that feature. Meanwhile, each leaf node represents a classification label or regression value.

The structure of the DTSVM algorithm model is similar to a decision tree, where nodes represent decision or split points, and leaf nodes indicate the final classification results. Due to the specifications outlined in international standard 'ISO 139-1973 Textiles—Standard atmospheres for conditioning and testing' regarding the environmental temperature and humidity range in experiments, threshold detectors are employed at these nodes for classification. Simultaneously, the behavior state of the operating personnel is a simple boolean value, hence threshold detectors are also utilized for classification. As for the electrical parameters, support vector machines (SVM) are employed at nodes or leaf nodes to conduct more refined data classification, enhancing the model's expressive capability, particularly when dealing with complex decision boundaries.

In the process of constructing a decision tree, the algorithm selects the optimal features and split points based on relevant criteria, aiming to make the data as pure as possible in each branch (i.e., samples of the same category are clustered together as much as possible). Information Gain (ID3) is an algorithm used to measure the contribution of features to the decision tree. Given a set $D$, the proportion of samples for the i-th class in the set is $p_i$(i = 1, 2, . . . . . ., n), and the entropy $E(D)$ is represented as shown in Eq (9):

$$E(D) = -\sum_{i=1}^{n} p_i \log_2(p_i) \tag{9}$$

Assuming $m$ is the outcome of discrete attribute $b$, after splitting the set based on attribute $b$, $m$ nodes are formed, and the m-th node is labeled as $D^m$. The information gain $G = (D, b)$ from partitioning set $D$ based on attribute $m$ is as shown in Eq (10):

$$G(D, b) = E(D) - \sum_{m-1}^{M} \frac{|D^m|}{|D|} E(D^m) \tag{10}$$

Whereas, the larger is $G = (D, b)$, the better the effectiveness of the attribute partitioning.

**Support vector machine method.** Support Vector Machine (SVM) is a supervised binary classification model that excels in datasets with complex decision boundaries. The core idea of SVM is to find an optimal hyperplane that effectively separates data points of different classes while maximizing the margin, which is the minimum distance between the two categories. SVM performs exceptionally well in datasets with complex decision boundaries.

$z_i \in \{-1, 1\}$ represents the classification label, and the discriminant function is defined as shown in Eq (11):

$$z_i[(v \cdot x_i) + y] - 1 \geq 0, \quad i = 1, 2, \ldots, n \tag{11}$$

The optimization problem is solved using the Lagrange multiplier method, and the optimal decision function is as shown in Eq (12):

$$f(x) = \mathrm{sgn}(\sum_{i=1}^{N} a_i z(x_i \cdot x) + y) \tag{12}$$

Whereas, $N$ is the number of support vectors, and $a_i$ is the Lagrange multiplier.

If the slack variable $\xi_i \geq 0$ is added to the previous Eq, it can be reformulated as shown in Eq (13):

$$z_i[(v \cdot x_i) + y] - 1 - \xi_i \leq 0, \quad i = 1, 2, \ldots, n \tag{13}$$

Minimizing the objective function using quadratic programming, i.e. as shown in Eq (14),

$$minQ(v, \xi) = \frac{1}{2} \| \omega\|^2 + C(\sum_{i-1}^{n} \xi_i) \tag{14}$$

The parameter $C$ in the previous Eq represents the penalty factor. The minimized objective function is transformed into a linear problem through nonlinear knowledge, resulting in the optimal classification plane and the expression of the optimal classification function as follows

Eq (15):

$$f(x) = sgn(\sum_{i-1}^{N} a_i * z_i K(x_i, x) + y^*) \tag{15}$$

In the previous Eq, $K(x_i, x)$ is the kernel function, and the kernel function must satisfy the requirements of Mercer's theorem. The choice of kernel function in support vector machines will affect the classification results. Electrical parameters include multiple dimensions: current, voltage, power. SVM is effective in classifying high-dimensional spaces, maintaining good performance even in cases of high-dimensional feature spaces. Polynomial kernel functions, radial basis kernel functions, and linear kernel functions are commonly used as kernel functions in support vector machines. The radial basis kernel function is defined as shown in Eq (16):

$$K(x_i, x) = exp(-\eta \parallel x_i - x \parallel^2) \tag{16}$$

The kernel equation $\eta$ and the penalty factor $C$ are two key parameters that can impact the performance of support vector machine classification. Therefore, it is necessary to find a balance between achieving the goals of 'correct classification of training samples' and 'maximizing the margin of the decision function' by adjusting the penalty factor C and parameters of the kernel function. This ensures optimal performance of the model.

Ten-fold cross-validation can be employed to assess the classification performance of SVM by randomly partitioning the training samples of multi-source data into 10 mutually exclusive subsets$\{S_1, S_2, \ldots, S_{10}\}$, representing ten folds. Each fold is of approximately equal size, and the process is iterated 10 times. In the j-th iteration, the $S_j$subset is chosen as the test set, while the sum of the remaining subsets forms the training set. The decision function is trained on the training set and then tested on the corresponding test set. The number of misclassified samples in each iteration is denoted as $I_j$. After 10 iterations, the ten-fold cross-validation error $\{I_1, I_2, \ldots, I_{10}\}$ is calculated as the ratio of the sum of misclassified samples over the total number of samples. The formula is as shown in Eq (17):

$$e = \frac{\sum_{j=1}^{k} I_j}{\Sigma_{j=1}^{k}(l_j/n)} \tag{17}$$

In the equations: $e$ represents the estimated error rate; $k$ is the iteration count, where in the context of ten-fold cross-validation $k$ equals 10; $n$ is the total number of samples. Based on the cross-validation error, optimal SVM hyperparameters can be selected, not only to improve the classification performance of SVM but also to prevent overfitting. The introduction of ten-fold cross-validation for training SVM involves randomly dividing the training set into 10 subsets. In each iteration, 9 subsets are used for training, and the remaining subset is used for validation. The average of the results from 10 cross-validation iterations is then used to evaluate the algorithm's accuracy. Its characteristic lies in the ability to repeatedly use randomly generated sub-samples for training and validation. Each training result is validated once, contributing to improving model accuracy and generalization ability.

## Experimental design

This section aims to explore the implementation process of the situational awareness experiment for fabric tear performance based on multi-source data. The purpose of this experiment is to achieve real-time monitoring of fabric sample states and traceable data status through the collection and analysis of data from different sensors and devices. The section begins with the on-site setup of SASFTPD-MSD and proceeds to conduct situational awareness experiments

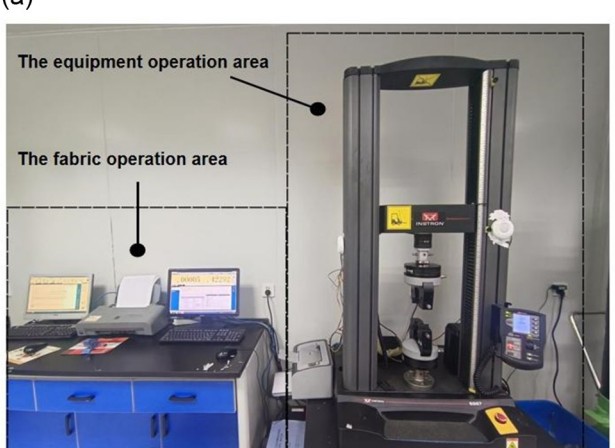
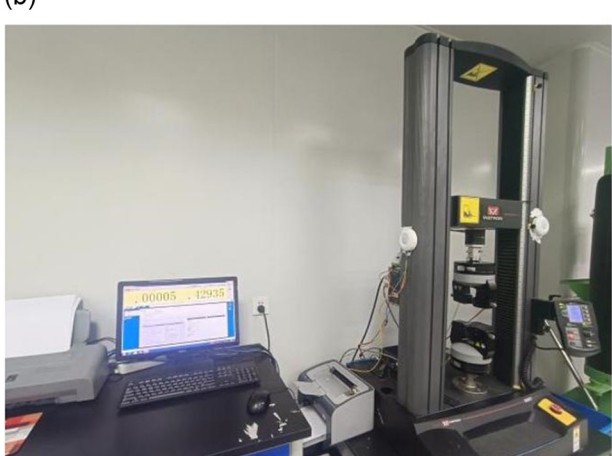

**Fig 6.** Laboratory Site Layout: (a) Main view; (b)side view. This is the laboratory Site Layout. (a) is the main view. (b) is a side view. It is divided into two areas: fabric operation area and equipment operation area.

based on multi-source data for different fabrics. Through the analysis and processing of the data collected on-site, real-time monitoring of fabric sample states and traceable data status can be achieved.

## On-site setup of SASFTPD-MSD

This paper relies on SASFTPD-MSD to collect multi-source data. The diverse data originates from various sensors strategically positioned around the working equipment in fabric tear performance testing experiments. The on-site arrangement is illustrated in Fig 6(a) and 6(b), (a) is Main view, (b) is side view. It is divided into the fabric operation area and the equipment operation area. The fabric operation area is responsible for preparing samples and printing archiving data records at the end of each experiment. The equipment operation area is responsible for operating the equipment during experiments.

CRE constant rate elongation tester uses the Instron 5967 dual-column tabletop testing machine, and the equipment parameters are shown in Table 3. The laboratory temperature and humidity sensor in the system is a temperature and humidity sensor with an accuracy of 0.1. The equipment and fabric operator behavior detection device use a 5.8GHz millimeter-wave radar human presence sensor. This sensor is an active detection sensor that can detect human heartbeat, respiration, micro-movement, and motion. The power parameter detector uses a Power Distribution Unit (PDU) with voltage, current, and power detection capabilities,

**Table 3. Equipment parameters of Instron 5967 dual-column tabletop testing machine.**

| Parameter | Unit | Scope | Parameter | Unit | Scope |
|---|---|---|---|---|---|
| Load capacity | KN | 30 | Return speed | mm/min | 1016 |
| Test accuracy | % | ≤0.01 | Speed accuracy | % | ≤0.5 |
| Displacement measurement accuracy | % | ≤0.5 | Stretch speed | mm/min | 100±1 |
| Gauge length | mm | 100±1 | Data collection frequency | times/s | 8 |
| Power requirements | VA | 900 | Power | KW | 1.6 |

**Table 4. Sensors and acquisition frequencies.**

| Name in system | Sensor type | Measuring range | Resolution | Sampling frequency |
|---|---|---|---|---|
| Laboratory temperature and humidity sensor | Temperature and humidity sensor | -40˚C ∼ +125˚C | 0.1˚C | 6 times/min |
| | | 0%RH ∼ 100%RH | 0.1%RH | |
| Equipment and fabric operator behavior detectors | 5.8G millimeter wave radar human presence sensor | Motion sensing distance ≤5m There is a sensing distance ≤3m | | 3 times/min |
| Electrical parameter detector | PDU | ≤2500W | 1W | 20Hz |
| | | ≤10A | 0.01A | |
| | | 250V AC | 0.01V | |

which can monitor voltage, current, and power parameters in real-time. The specific sensor names and acquisition frequencies are shown in Table 4.

## Situational awareness experiment based on multi-source data

The experiment begins by conducting tear performance tests on two sets of trouser-shaped fabrics, one made of polyester fiber and the other of blended fibers (nylon, spandex, lyocell). Each set consists of 5 samples in both the warp and weft directions. Subsequently, four additional fabric experiments are conducted after the blended fiber experiment, involving cotton, wool, viscose fiber, and another blend (polyester, regenerated cellulose, spandex). The details of the fabrics and their specific properties are outlined in Table 5:

**Data feature analysis.** Sample Preparation and Humidity Adjustment: On the preceding day, samples are prepared with a preparation time of approximately 15 minutes per set, following the personal operating habits of the operator. These samples are then placed in a standard atmospheric environment for humidity adjustment over a 24-hour period. The experiment is conducted the next day, and no monitoring is performed during this stage of the experiment. Experiment Execution: The experiment begins with the startup of the equipment and preparation for testing. After a preloading period of 30 minutes, the operations commence.

Initially, a tensile test is conducted on polyester fiber, followed by a blended fiber (nylon, spandex, lyocell) tensile test. Each fabric type comprises 10 specimens, with 5 samples taken in the warp direction and 5 in the weft direction.

For the collected two sets of multi-source data, data preprocessing was performed, and the preprocessed data (excluding data normalization) was visualized.

The multi-dimensional sensor recorded the data obtained by SASFTPD-MSD throughout the entire experiment, documenting the working states of two sets of experiments involving polyester fiber and blended fabrics (nylon, spandex, Lyocell), as shown in Fig 7, The

**Table 5. Fabrics and their properties.**

| Fabric | Performance |
|---|---|
| Polyester | Almost inelastic |
| Blended fabrics (nylon, spandex, lyocell) | Good elasticity |
| Cotton | Almost inelastic |
| Wool | Good elasticity |
| Viscose fiber | Medium elasticity |
| Blended(polyester, regenerated cellulose, spandex) | Good elasticity |

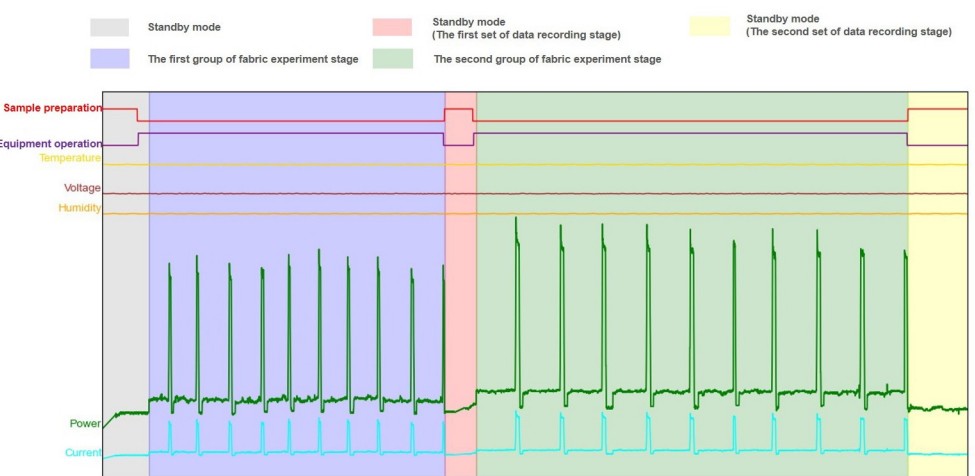

**Fig 7. Time-series state diagram of multi-dimensional perception data.** The SASFTPD-MSD collects multiple sources of data after the device is powered on, including temperature, humidity, voltage, operator status changes, and current and power changes.

SASFTPD-MSD collects multi-source data after the device is powered on, During this experiment, temperature, humidity, and voltage smoothly fluctuated within reasonable ranges. The variation in the operator's state, obtained from the detection radar, reflected the working state of the experiment when one operator was performing tasks. The changes in current and power reflected the equipment's states, including standby, preloading, tearing and stretching, and stop-reset. The standby state involves the initiation of the equipment and the preloading process. During this phase, the equipment power gradually rises from 0 to the standby power. The equipment remains in standby mode until the tearing and stretching experiment is initiated. The first set of experiments involves polyester fiber fabric, with 10 sample groups. From the power data, it is evident that the power during tearing and stretching fluctuates around 100W, while the power required for resetting after each tearing is approximately 300W, lasting for about 10 seconds. Throughout this process, the operator remains near the equipment. After completing the experiment, the equipment returns to standby mode, and the operator performs data operations on the computer at the operating console. The experimental process for the second set of blended fabric (nylon, spandex, Lyocell) samples is similar to the first set of polyester fiber fabric, but it is noticeable that the stretching time and reset power are significantly higher than those in the first set, as shown in Fig 8. The 10 sets of data in the experimental process consist of 5 sets of tearing experiments in the radial direction and 5 sets in the weft direction for each type of fabric. We compared the mechanical time series charts with the system-perceived electrical power time series data, as shown in Figs 9a–9d and 10a–10d. Through the comparison of the experiments, it can be observed that due to the differential mechanical characteristics of tearing in the radial and weft directions of the fabric, fabrics with different elasticity also exhibit differential mechanical characteristics during the tearing experiment. By comparing electrical power with mechanical characteristics, the consistent fluctuation trends of the two indicate that we can use fewer power perception data to reflect the mechanical changes. This demonstrates that the SASFTPD-MSD system can effectively restore the working process.

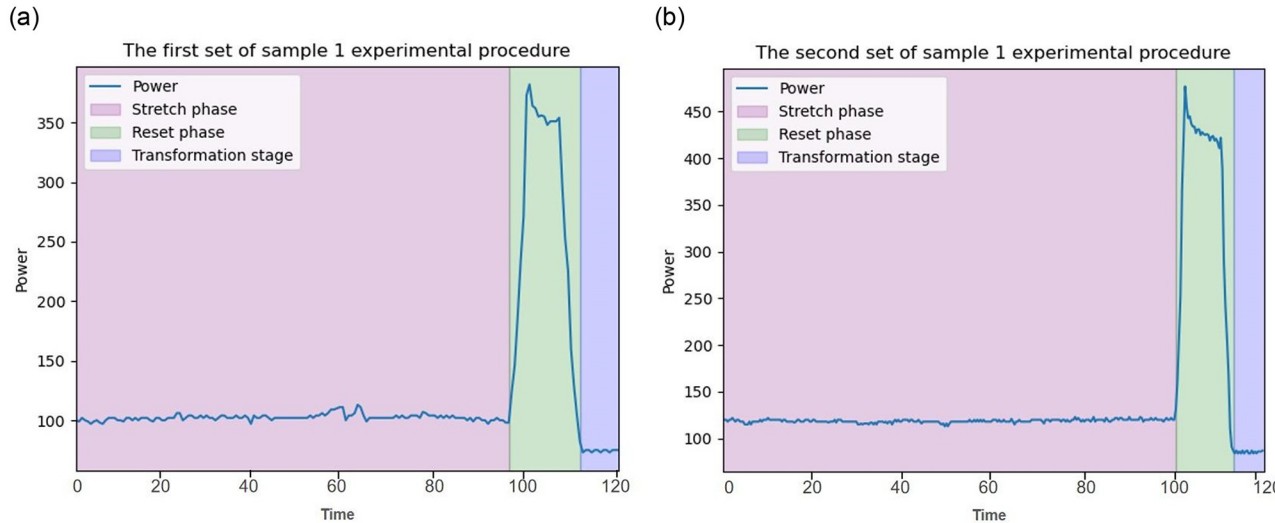

**Fig 8.** Single tearing experiment power timing chart: (a) Polyester Fiber Fabricl; (b) Blended (Nylon, Spandex, Lyocell) Fabric. This is single tearing experiment power timing chart. (a) is Polyester Fiber Fabricl tearing experiment power timing chart. (b) is Blended (Nylon, Spandex, Lyocell) Fabric tearing experiment power timing chart. They reflect the three working states of the equipment operation, which are: stretch state, reset state and change state.

## Experimental results and discussion

### Experimental results

In the dataset for the two sets of experiments mentioned above, there are fewer instances of anomalous data compared to normal data. To address this imbalance, data augmentation is applied to the anomalous data. Given that the dataset is a time series, a sliding window approach is used to generate new data points. The augmented multi-source dataset is then split into a training set (7/10 of the data) and a testing set (3/10 of the data). A device working state classification is conducted based on the DTSVM method, categorizing the data into six different working states, as illustrated in Table 6:

To better illustrate the classification performance of the model, a confusion matrix is introduced to analyze the results. The main diagonal elements represent the number of samples correctly identified for each working state, while the off-diagonal elements represent the number of recognition errors. The classification results for two sets of materials are shown in Fig 11a and 11b, with the model achieving accuracy rates of 98.88% and 99.10% for the respective materials. The corresponding generalization errors, calculated after ten-fold cross-validation, are 0.0195 and 0.0052. Through the confusion matrix, the following indicators can be obtained: TP-True Positive (samples are true, predicted as true), FP-False Positive (samples are false, predicted as true), TN-True Negative (samples are false, predicted as false), FN-False Negative (samples are true, predicted as false).

Overall Accuracy is expressed by Eq (18):

$$\mathrm{OverallAccuracy} = \frac{\mathrm{TP} + \mathrm{TN}}{\mathrm{TP} + \mathrm{TN} + \mathrm{FP} + \mathrm{FN}} \times 100\% \tag{18}$$

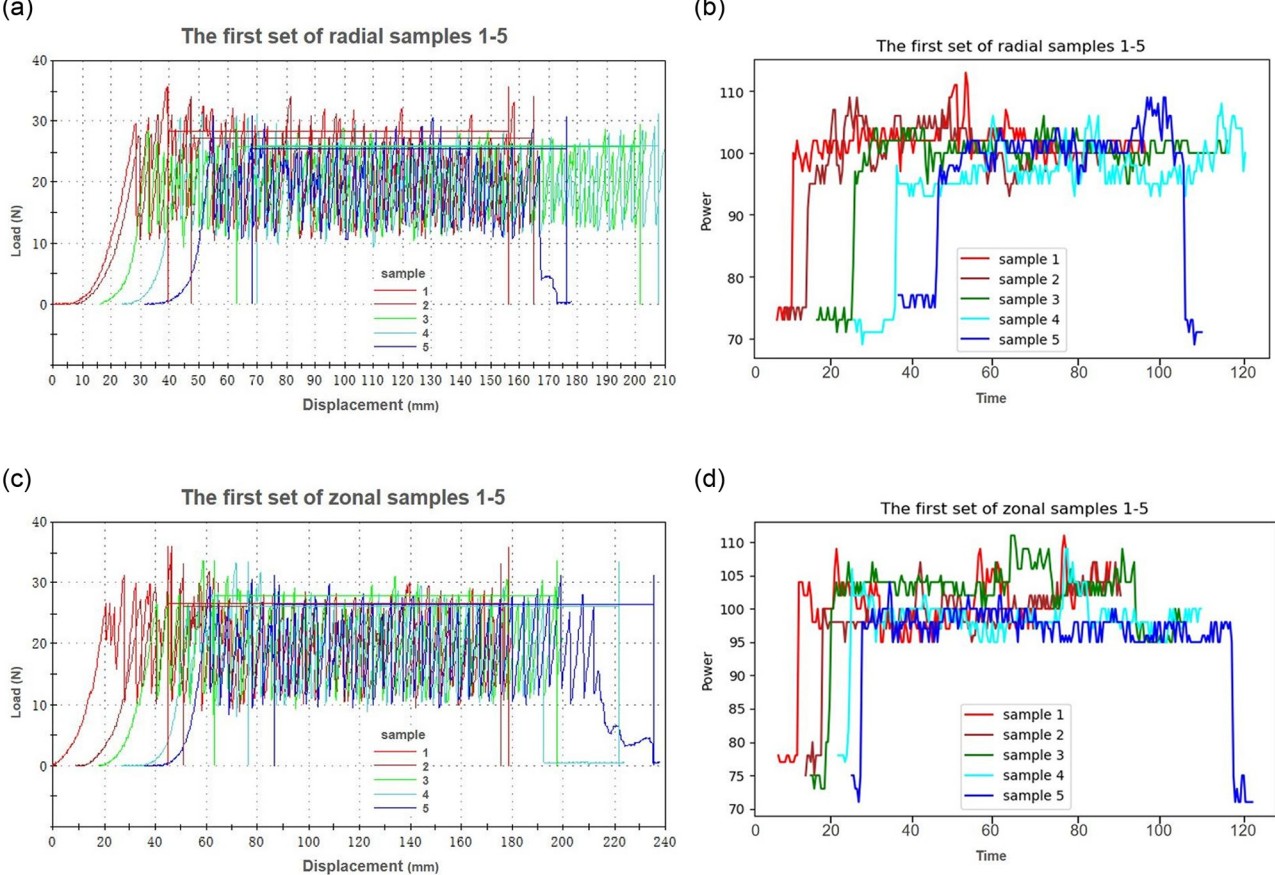

**Fig 9.** Comparison Chart of Polyester Fiber Fabricss: (a) Radial Mechanical Time Series Chart; (b) Radial Electrical Power Time Series; (c)Weft-wise Mechanical Time Series Chart; (d)Weft-wise Electrical Power Time Series Chart. By comparing the warp and weft mechanical time series diagram of polyester fiber fabric with the warp and weft system sensing electrical power time series data diagram, the fluctuation trend of the two is consistent.

Micro-Precision is expressed by Eq (19):

$$\text{Micro} - \text{Precision} = \text{Micro} - \text{Precision} = \frac{\sum_{k=1}^{C} \text{TP}_k}{\sum_{k=1}^{C} \text{TP}_k + \sum_{k=1}^{C} \text{FP}_k} \tag{19}$$

Micro-Recall is expressed by Eq (20):

$$\text{Micro} - \text{Recall} = \text{Micro} - \text{Precision} = \frac{\sum_{k=1}^{C} \text{TP}_k}{\sum_{k=1}^{C} \text{TP}_k + \sum_{k=1}^{C} \text{FN}_k} \tag{20}$$

$C$ represents the number of categories, and $TP_k$, $FP_k$, and $FN_k$ denote the number of true, false positive, and false negative cases in category $k$, respectively.

Micro-F1 Score is the reconciled average of micro-averaged precision and recall and is expressed by Eq (21),

$$\text{Micro} - \text{F1Score} = \text{Micro} - \text{F1} = \frac{2 \cdot \text{Micro} - \text{Precision} \cdot \text{Micro} - \text{Recall}}{\text{Micro} - \text{Precision} + \text{Micro} - \text{Recall}} \tag{21}$$

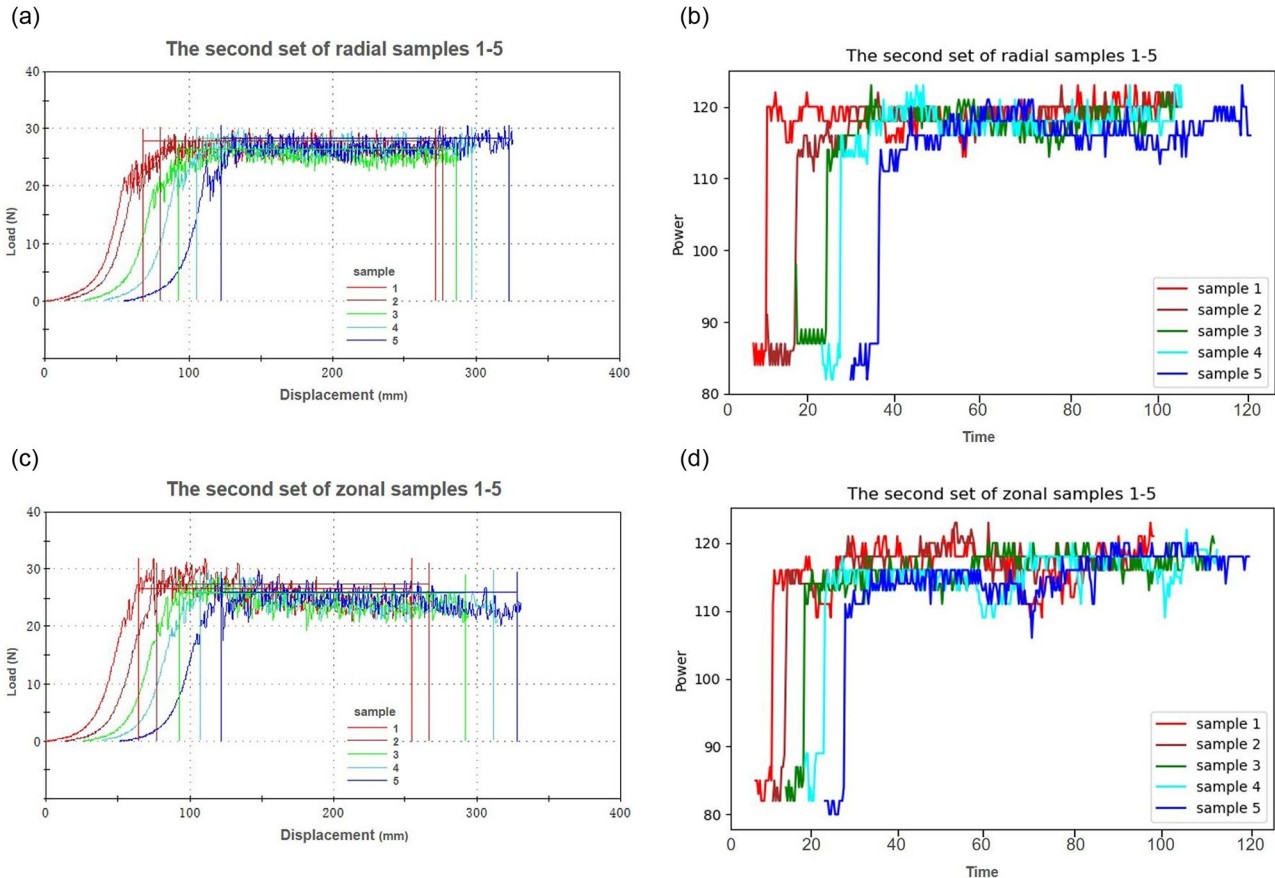

**Fig 10.** Comparison of Blended (Nylon, Spandex, Tencel) Fabrics: (a) Radial Mechanical Time Series Chart; (b)Radial Electrical Power Time Series Chart; (c)Weft Mechanical Time Series Chart; (d)Weft Electrical Power Time Series Chart. By comparing the warp and weft mechanical time series diagram of blended (Nylon, Spandex, Tencel) fabrics with the warp and weft system sensing electrical power time series data diagram, the fluctuation trend of the two is consistent.

From the formulas for Overall Accuracy, Micro-Precision, Micro-Recall and Micro-F1 Score: Overall Accuracy = Micro-Precision = Micro-Recall = Micro-F1 Score.

The ROC curve is shown in Fig 12, which is used to evaluate the performance of the classifier. The Receiver Operating Characteristic (ROC) curve is a comprehensive indicator reflecting both recall and false positive rate. The vertical axis represents recall, and the horizontal axis

**Table 6. Classification Categories and States.**

| Category | Status |
|---|---|
| The Class 1 | Non-test environment temperature and humidity status |
| The Class 2 | Non-test running status |
| The Class 3 | No one is present during stretching |
| The Class 4 | Someone is present during stretching |
| The Class 5 | Someone is preparing the fabric |
| The Class 6 | No one is preparing the fabric |

(a)                                                     (b)

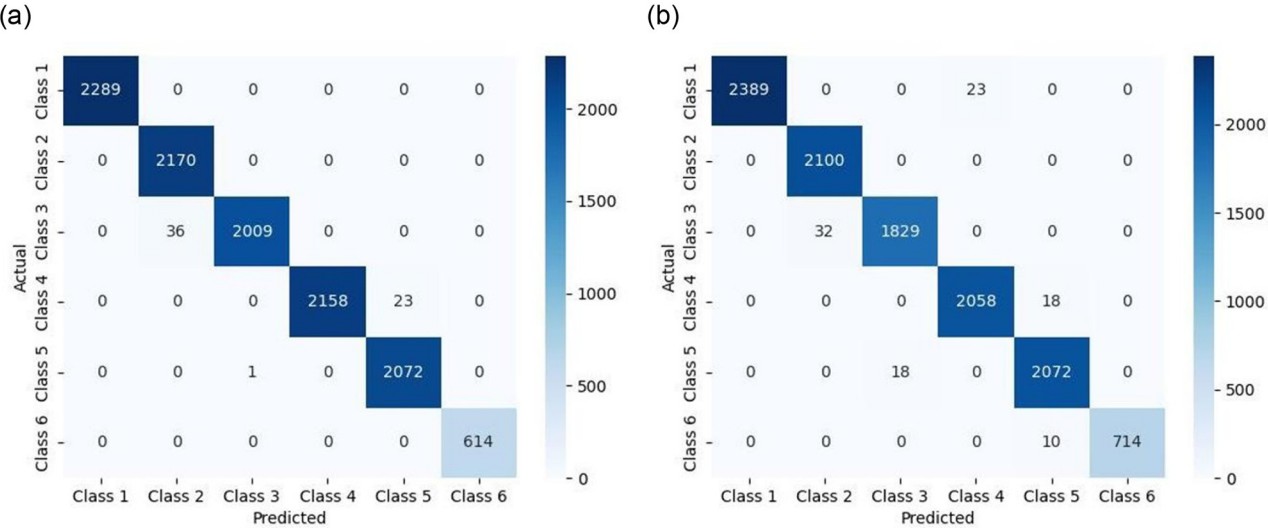

**Fig 11.** Confusion Matrix: (a) Polyester Fiber Fabricl; (b) Blended (Nylon, Spandex, Lyocell) Fabric. The accuracy of the DTSVM model for (a) Polyester Fiber Fabricl and (b) Blended (Nylon, Spandex, Lyocell) fabrics reached 98.88% and 99.10%, respectively, as represented by the confusion matrix visualization.

represents the false positive rate (FPR), i.e. as shown in Eq (22),

$$\text{FPR} = \frac{\text{FP}}{\text{FP} + \text{TN}} \qquad (22)$$

The area under the ROC curve, denoted as Area Under the Curve (AUC), represents the performance of the classifier. Typically, a higher AUC value, closer to 1, indicates better classifier performance.

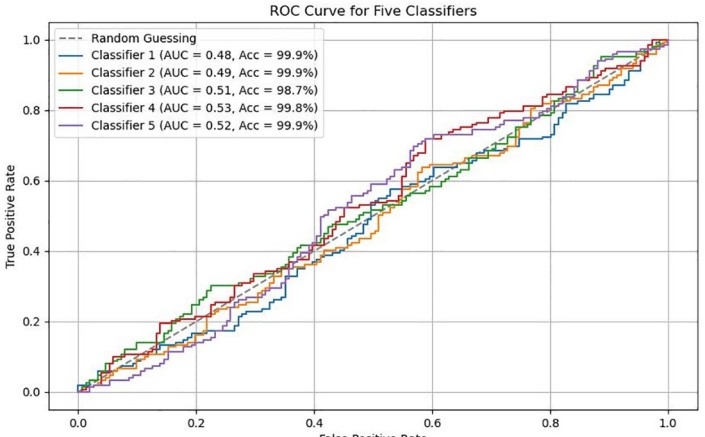

**Fig 12. ROC curve.** The ROC curve is used to evaluate the performance of the classifier, the area under the ROC curve, denoted as Area Under the Curve (AUC), represents the performance of the classifier.

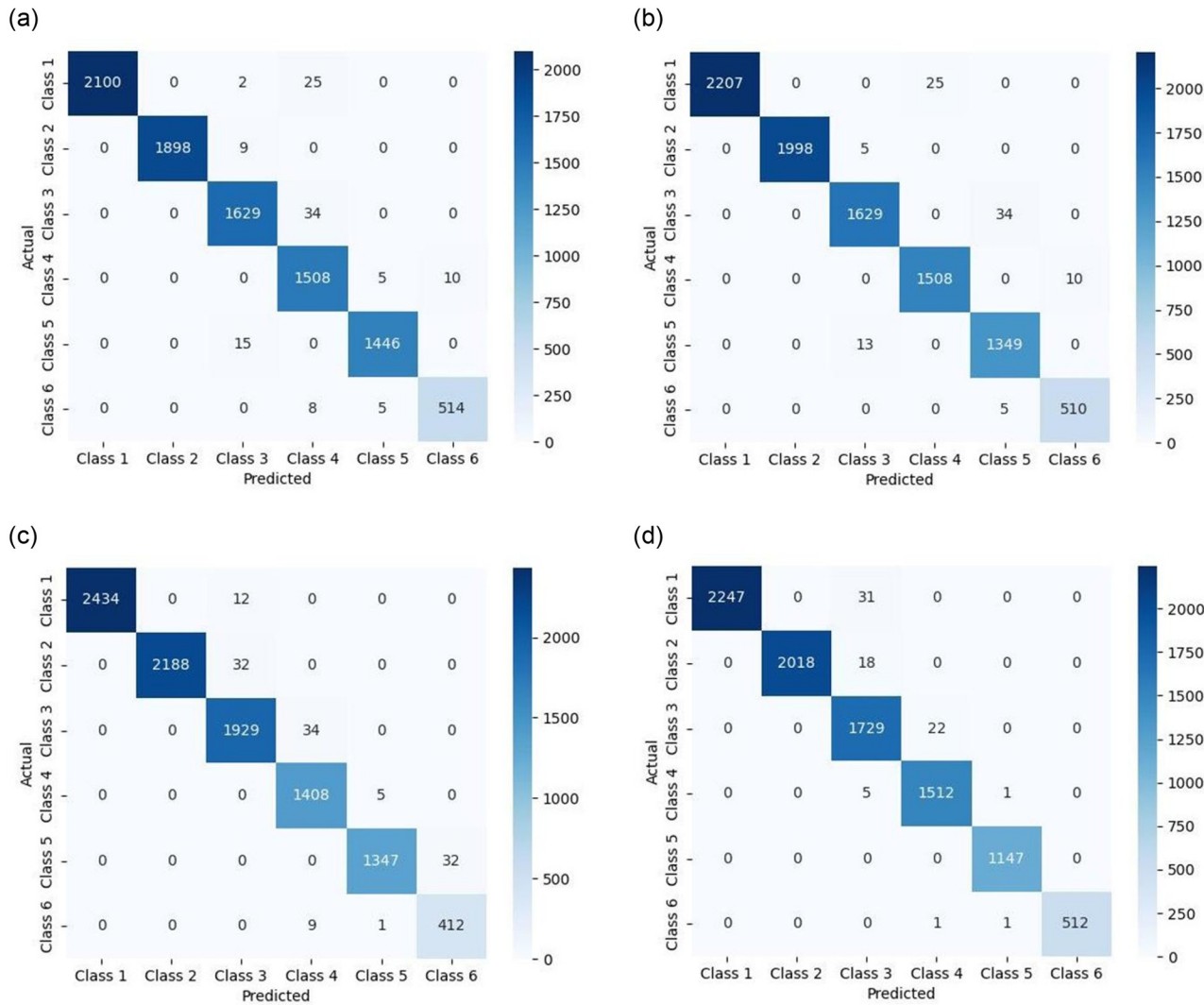

**Fig 13.** Visualization of Confusion Matrices for Four Groups of Classification Results: (a) Cotton; (b)Wool; (c)Viscose fiber; (d)Blended (polyester, regenerated cellulose, spandex). The accuracy of the DTSVM model for (a) Cotton, (b)Wool, (c)Viscose fiber, and (d)Blended (polyester, regenerated cellulose, spandex) reached 98.77%, 99.01%, 98.73%, 99.15%, respectively, as represented by the confusion matrix visualization.

Regarding the experimental data of cotton, wool, viscose fiber, and blended fabrics (polyester, regenerated cellulose, spandex), multi-classification was conducted for the different states. The confusion matrices of the classification results are shown in Fig 13a–13d):

Experimental comparisons with other classification methods (one-against-all (OAA) and one-against-one (OAO)) are performed along with DTSVM classification experiments. The accuracy of multi-classification is obtained as shown in Table 7. Upon comparison, the classification accuracy of the DTSVM model proposed in this paper is much higher than other methods.

## Experimental discussion

The SASFTPD-MSD constructed in this paper collects multi-source data from multiple sensors. Through preprocessing steps such as data denoising and alignment, the DTSVM

**Table 7. Multi-classification accuracy for different fabrics.**

| Fabric | DTSVM Overall Accuracy | OAO Accuracy | OAA Accuracy |
|---|---|---|---|
| Cotton | **98.77%** | 89.71% | 91.21% |
| Wool | **99.01%** | 90.21% | 92.72% |
| Viscose fiber | **98.73%** | 89.35% | 91.14% |
| Blended (polyester, regenerated cellulose, spandex) | **99.15%** | 90.13% | 92.83% |
| Polyester | **98.88%** | 89.87% | 91.05% |
| Blended fabrics (nylon, spandex, lyocell) | **99.10%** | 90.11% | 92.76% |

algorithm model is applied to address the multi-classification problem of device operation states in fabric tear performance detection. For experiments conducted on six different elastic fabric sets, the multi-class state accuracy of the constructed SASFTPD-MSD consistently exceeds 98.73%. The introduction of ten-fold cross-validation enhances the model's generalization ability.

## Conclusion and outlook

### Conclusion

The proposed SASFTPD-MSD system integrates various sensors and employs the DTSVM algorithm for multi-classification of device states, enabling real-time monitoring of the fabric tearing performance experimental process. Experimental results indicate that the system can effectively detect the fabric tearing performance process with high accuracy and reliability. The study utilizes multiple sensors to capture multidimensional data during the fabric tearing performance experiments, including temperature, humidity, operator behavior states, and electrical parameter information. Data preprocessing is also performed to ensure the quality and reliability of the data. Specifically, a state classification method based on DTSVM was employed. Ten-fold cross-validation was introduced for training, and the model's classification performance was tested using a test set to evaluate its accuracy. The experimental results show that the method can solve the challenges of multi-classification in equipment status and data traceability in the inspection process, especially in equipment status classification, which exhibits a high accuracy of more than 98.73%. The widespread application of the system contributes to the continuous improvement of the workflow and traceability of fabric tearing performance testing processes. It also provides new ideas and solutions for the application of intelligent perception technology in the field of traditional textile inspection.

### Outlook

As time progresses, the performance and technical standards of textiles may undergo changes. Therefore, it is essential to regularly update and maintain datasets and models to ensure their accuracy and reliability. In the context of this paper, SASFTPD-MSD has only performed multi-classification of equipment operating states, without distinguishing abnormal experimental results that may occur during the experimental process. Consequently, in future work, real-time monitoring and prediction of abnormal experimental results will be implemented to enhance the accuracy of experiments.

## Acknowledgments

We would like to thank Qiyu Chen, Ding Kai, Xiaohu Qian and Hangping Cao for their help in collecting the data as well as all the volunteers who participated in our experiments. We would also like to thank Xu Yiping for her suggestions on modifying the fabric properties.

## Author Contributions

**Conceptualization:** Jianmin Huang.

**Data curation:** Yifan Zhang.

**Formal analysis:** Qingchun Jiao, Gaoqing Xu.

**Funding acquisition:** Jianmin Huang.

**Investigation:** Qingchun Jiao.

**Methodology:** Qingchun Jiao.

**Project administration:** Jianmin Huang, Gaoqing Xu.

**Resources:** Jianmin Huang, Gaoqing Xu, Lijun Wang.

**Software:** Lijun Wang.

**Supervision:** Qingchun Jiao, Dong Yue.

**Validation:** Lijun Wang.

**Visualization:** Yifan Zhang.

**Writing – original draft:** Jianmin Huang, Qingchun Jiao, Yifan Zhang.

**Writing – review & editing:** Jianmin Huang, Qingchun Jiao, Yifan Zhang.

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
