## [Decision Letter · Decision Letter 0]

10 Mar 2024

PONE-D-24-02493Fabric tearing performance state perception and classification driven by multi-source dataPLOS ONE

Dear Dr. Jiao,

Thank you for submitting your manuscript to PLOS ONE. After careful consideration, we feel that it has merit but does not fully meet PLOS ONE’s publication criteria as it currently stands. Therefore, we invite you to submit a revised version of the manuscript that addresses the points raised during the review process.

We look forward to receiving your revised manuscript.

Kind regards,

Praveen Kumar Donta, Ph.D.

Academic Editor

PLOS ONE

3. In this instance it seems there may be acceptable restrictions in place that prevent the public sharing of your minimal data. However, in line with our goal of ensuring long-term data availability to all interested researchers, PLOS’ Data Policy states that authors cannot be the sole named individuals responsible for ensuring data access (http://journals.plos.org/plosone/s/data-availability#loc-acceptable-data-sharing-methods).

Reviewers' comments:

Reviewer's Responses to Questions

**Comments to the Author**

1. Is the manuscript technically sound, and do the data support the conclusions?

Reviewer #1: Yes

Reviewer #2: Yes

Reviewer #3: Yes

2. Has the statistical analysis been performed appropriately and rigorously? 

Reviewer #1: Yes

Reviewer #2: No

Reviewer #3: Yes

3. Have the authors made all data underlying the findings in their manuscript fully available?

Reviewer #1: Yes

Reviewer #2: Yes

Reviewer #3: No

4. Is the manuscript presented in an intelligible fashion and written in standard English?

Reviewer #1: Yes

Reviewer #2: No

Reviewer #3: Yes

5. Review Comments to the Author

Reviewer #1: In this manuscript, the authors propose a state-awareness and classification approach, Decision Tree Support Vector Machine (DTSVM), for fabric tear performance testing based on multi-source data. The paper's structure seems to be appropriate and aligned with a scientific article's objectives. The following opinions were formed by examining the article in detail:

The texts in some figures are not in a readable state. Authors should also make sure that the texts can be read if printed on paper.

The proposed method's F-score, precision, and recall values should also be given alongside accuracy to provide a more comprehensive evaluation of a model's performance.

The quantitative results obtained from the study should be added to the Conclusion of the paper.

Reviewer #2: The manuscript discusses an important issue in the textiles industry concerning the tensile strength of materials. The proposed approach utilizes multi-source data to enhance state-awareness and categorization, providing a unique solution to minimize the impact of different elements in fabric tear performance testing. Overall, the manuscript provides valuable insights into improving the testing process. However, a few minor revisions are recommended to enhance clarity and address specific points.

- The introduction and conclusion should be revised to concisely communicate the importance of the suggested approach and its potential impact on fabric tear performance testing.

- The technique section provides a thorough description of the methodical design and implementation of Decision Tree Support Vector Machine (DTSVM) for classification. However, more details on the preprocessing steps and the rationale behind choosing DTSVM over other classification methods would enhance the comprehensibility of the methodology.

- The experimental results and discussion are effectively presented, demonstrating the efficacy of the proposed system. However, it would be beneficial to include a more detailed discussion on the implications of achieving an accuracy, especially in comparison to existing methods or industry standards. Also, addressing potential limitations or challenges in implementing the proposed approach would provide a more comprehensive perspective.

- The manuscript is generally well-written, but some sentences could be revised for improved clarity and flow. Minor grammatical and typographical errors should be corrected throughout the manuscript for example no space between sentences in page 5, line 176.

Reviewer #3: The manuscript titled "Fabric tearing performance state perception and classification driven by multi-source data" by Jianmin Huang et al. introduces a novel approach to enhance the reliability and traceability of fabric tear performance testing through the use of a multi-source data-driven Decision Tree Support Vector Machine (DTSVM). This methodological innovation is particularly timely and significant, given the increasing complexity of textile materials and the demand for precise and reliable testing methods in the textile industry. The manuscript is well-organized, with a clear exposition of the problem statement, methodology, experimental setup, and findings. However, there are areas where the manuscript could be improved to enhance its clarity, completeness, and impact.

Comments:

1)The manuscript successfully demonstrates the application and effectiveness of the DTSVM algorithm in classifying the states of fabric tear performance testing with impressive accuracy. Integrating multi-source data, including electrical parameters, environmental conditions, and operator behaviour, is innovative and addresses a gap in existing testing protocols.

2) It would be beneficial to further elucidate the algorithm's novelty compared to other state-of-the-art methods in textile testing. A more detailed discussion of the decision-making process of the DTSVM and its advantages over traditional methods could strengthen the manuscript.

3)The experimental setup and the data analysis are thoroughly described, providing a solid foundation for the claims made regarding the system's accuracy and reliability. Including ten-fold cross-validation is commendable, as it enhances the credibility of the classification performance.

4)Additional information on the selection process for the fabrics tested would enhance the reader's understanding of the system's applicability to a wide range of textile materials.

5)The manuscript outlines the potential impact of the proposed system on improving the workflow and traceability of fabric tear performance testing processes. However, the discussion on practical implications is limited.

6)It would be valuable to expand on how this system can be integrated into existing textile testing laboratories and its compatibility with current industry standards. Additionally, outlining future research directions, especially concerning the system's adaptability to new materials and testing conditions, would be beneficial.

7)While the manuscript is generally well-structured, some sections could benefit further clarification and refinement. For instance, the introduction of the DTSVM algorithm and its underlying principles could be made more accessible to readers not familiar with machine learning techniques.

The manuscript presents a significant contribution to textile testing, offering a robust method for the real-time monitoring and classifying of fabric tear performance states. With some revisions, particularly in enhancing the discussion on the system's practical implications and future work, the manuscript could provide a valuable resource for researchers and practitioners in the textile industry.

6. PLOS authors have the option to publish the peer review history of their article (what does this mean?). If published, this will include your full peer review and any attached files.

Reviewer #1: No

Reviewer #2: No

Reviewer #3: No

---

## [Author Response · Author response to Decision Letter 0]

18 Mar 2024

Dear Editors and Reviewers,

We are grateful for the opportunity to revise our manuscript entitled "Fabric tearing performance state perception and classification driven by multi-source data". Thank you very much for your comments and professional advice. These opinions help to improve academic rigor of our article. Based on your suggestion and request, we have made corrected modifications on the revised manuscript. Meanwhile, the manuscript was also professionally reviewed. We hope that our work can be improved again. Furthermore, we would like to show the details as follows:

Academic editor’s comments

1.Please ensure that your manuscript meets PLOS ONE's style requirements, including those for file naming.

Response:

I have carefully read and understood PLOS ONE's formatting requirements, including file naming conventions. Before submitting my manuscript, I have thoroughly checked and adjusted my manuscript against the relevant standards to ensure that all files comply with the style requirements of PLOS ONE. We have ensured that our manuscript meets PLOS ONE's style requirements. If there is any non-conformity, please also point it out, and I will make corresponding changes as soon as possible. Meanwhile, to facilitate the review and post-processing of the manuscript, I have named all the files appropriately according to PLOS ONE's guidelines and ensured that the information is clear and accurate. 

2.Please note that PLOS ONE has specific guidelines on code sharing for submissions in which author-generated code underpins the findings in the manuscript. In these cases, all author-generated code must be made available without restrictions upon publication of the work. 

Response:

I would like to confirm that in order to comply with PLOS ONE's policy on code sharing, we have uploaded all the necessary code, datasets and related materials generated in the course of this research work to the GitHub platform. These resources are publicly available at the following link: https://github.com/yuyuyu123YUYUYU/data.

In our GitHub repository, readers can access, view, download, and reuse this code without restriction, and reproduce the experimental results on their own according to the provided documentation. We strongly believe that this will greatly enhance the transparency and reproducibility of this study and follows PLOS ONE's principle that author-generated code should be shared openly after publication of the manuscript.

3.In this instance it seems there may be acceptable restrictions in place that prevent the public sharing of your minimal data. However, in line with our goal of ensuring long-term data availability to all interested researchers, PLOS’ Data Policy states that authors cannot be the sole named individuals responsible for ensuring data access.

Response:

I understand and respect the core PLOS principle of data sharing, which is to ensure that research data are accessible and reusable by other researchers over time. While we are currently unable to directly disclose the smallest datasets containing sensitive information due to certain reasonable restrictions, we have taken steps to maximize our ability to meet this policy requirement.

Specifically, we have stored datasets that have been processed as de-identified as possible in a controlled-access environment and made them available to vetted researchers through an appropriate data hosting organization or third-party platform. The address of this platform is listed below: https://github.com/yuyuyu123YUYUYU/data.

Researchers can gain legitimate access to the data they need by submitting a reasonable data use request and signing the necessary confidentiality agreement. This ensures compliance with data use, as well as data security and privacy protection.

We understand the importance of open science and are committed to promoting data availability and transparency to the greatest extent possible while meeting data protection regulations.

4.Please review your reference list to ensure that it is complete and correct. If you have cited papers that have been retracted, please include the rationale for doing so in the manuscript text, or remove these references and replace them with relevant current references. Any changes to the reference list should be mentioned in the rebuttal letter that accompanies your revised manuscript. If you need to cite a retracted article, indicate the article’s retracted status in the References list and also include a citation and full reference for the retraction notice.

Response:

After careful checking and verification, the reference list of the article is complete and accurate, and there are no references to papers that have been retracted.

Comments to the Author

1.Is the manuscript technically sound, and do the data support the conclusions?

Response:

I would like to express my deepest gratitude to you for the careful and professional technical review of our manuscript in your valuable time. In the course of our research, we followed a rigorous scientific method, and the data obtained were repeatedly verified and analyzed, aiming to truly and accurately reflect the experimental results and support our conclusions. Thank you for your recognition!

2.Has the statistical analysis been performed appropriately and rigorously?

Response:

In conducting this study, we have rigorously employed suitable data analysis methods to test our hypotheses and draw conclusions. We have exhaustively examined all the data and in the subsequent revision of the manuscript, we have added comparative experiments to verify the accuracy of the methodology proposed in this paper.

3.Have the authors made all data underlying the findings in their manuscript fully available?

Thank you very much for your review of this manuscript, especially for your interest in the openness of our data. We fully understand and support the principles of open access and data sharing to promote reproducibility and transparency in scientific research. In preparing this manuscript, we have ensured that all data supporting the research findings in this paper have been fully disclosed. In order to facilitate peer reproduction and validation of our findings, all necessary raw data have been stored in appropriate databases as required by the journal, with access details and links provided in the text.

4.Is the manuscript presented in an intelligible fashion and written in standard English?

Response:

Thank you for your approval, Regarding grammatical issues and typographical errors, we have engaged a proofreading agency and made changes to the text according to the proofreader's suggestions.

Reviewer 1#

In this manuscript, the authors propose a state-awareness and classification approach, Decision Tree Support Vector Machine (DTSVM), for fabric tear performance testing based on multi-source data. The paper structure seems to be appropriate and aligned with a scientific article objectives. The following opinions were formed by examining the article in detail: 

1.The texts in some figures are not in a readable state. Authors should also make sure that the texts can be read if printed on paper.

2.The proposed method; F-score, precision, and recall values should also be given alongside accuracy to provide a more comprehensive evaluation of a model; performance.

3.The quantitative results obtained from the study should be added to the Conclusion of the paper.

The author’s answer:

Thank you very much for your suggestions, which will be responded to line by line below:

1.In response to this issue, I have uploaded all of the images into the PACE website and have re-generated the image format to match your journal based on the advice given on the website, which has passed PACE's image inspection and is now repackaged in the file.

2.I have modified the formulas of Overall Accuracy, Micro-Average Precision, Micro-Average Recall, and Micro-F1 Score, and obtained the conclusion that the values of these items are the same through the formulas, and also added other classification methods (one-against-all (OAA) and one-against-one (OAO)) for experimental comparisons to conduct a more comprehensive evaluation of the model proposed in this paper, which is reflected in the chapter of "Experimental Results and Discussion".

3.I have added the quantitative results obtained in this study to the conclusion of the paper.

Reviewer 2#

The manuscript discusses an important issue in the textiles industry concerning the tensile strength of materials. The proposed approach utilizes multi-source data to enhance state-awareness and categorization, providing a unique solution to minimize the impact of different elements in fabric tear performance testing. Overall, the manuscript provides valuable insights into improving the testing process. However, a few minor revisions are recommended to enhance clarity and address specific points.

1.The technique section provides a thorough description of the methodical design and implementation of Decision Tree Support Vector Machine (DTSVM) for classification. However, more details on the preprocessing steps and the rationale behind choosing DTSVM over other classification methods would enhance the comprehensibility of the methodology.

2.The experimental results and discussion are effectively presented, demonstrating the efficacy of the proposed system. However, it would be beneficial to include a more detailed discussion on the implications of achieving an accuracy, especially in comparison to existing methods or industry standards. Also, addressing potential limitations or challenges in implementing the proposed approach would provide a more comprehensive perspective.

3.The manuscript is generally well-written, but some sentences could be revised for improved clarity and flow. Minor grammatical and typographical errors should be corrected throughout the manuscript for example no space between sentences in page 5, line 176.

4.The technique section provides a thorough description of the methodical design and implementation of Decision Tree Support Vector Machine (DTSVM) for classification. However, more details on the preprocessing steps and the rationale behind choosing DTSVM over other classification methods would enhance the comprehensibility of the methodology.

The author’s answer:

Thank you for recognizing the value of the research in this paper, and in response to your suggestion below is my response:

1.The importance and potential impact of the methodology has been added to the text, such as in the conclusion where it is stated that "It also provides new ideas and solutions for the application of intelligent perception technology in the field of traditional textile inspection."

2.This paper added other classification methods (one-against-all (OAA) and one-against-one (OAO)) for experimental comparisons to conduct a more comprehensive evaluation of the model proposed in this paper, which is reflected in the chapter of "Experimental Results and Discussion". comprehensive evaluation of the model proposed in this paper, which is reflected in the chapter of "Experimental Results and Discussion".

3.Thank you for your approval, Regarding grammatical issues and typographical errors, we have engaged a proofreading agency and made changes to the text according to the proofreader's suggestions.

4.About the details of the preprocessing has been added to the main text "Data Preprocessing", the formula will be embodied in the article, so that readers can easily understand the principle of DTSVM and why the choice of the algorithm, has been described in detail in the "Data Preprocessing".

Reviewer 3#

The manuscript titled "Fabric tearing performance state perception and classification driven by multi-source data" by Jianmin Huang et al. introduces a novel approach to enhance the reliability and traceability of fabric tear performance testing through the use of a multi-source data-driven Decision Tree Support Vector Machine (DTSVM). This methodological innovation is particularly timely and significant, given the increasing complexity of textile materials and the demand for precise and reliable testing methods in the textile industry. The manuscript is well-organized, with a clear exposition of the problem statement, methodology, experimental setup, and findings. However, there are areas where the manuscript could be improved to enhance its clarity, completeness, and impact.

1.The manuscript successfully demonstrates the application and effectiveness of the DTSVM algorithm in classifying the states of fabric tear performance testing with impressive accuracy. Integrating multi-source data, including electrical parameters, environmental conditions, and operator behaviour, is innovative and addresses a gap in existing testing protocols.

2.It would be beneficial to further elucidate the algorithm novelty compared to other state-of-the-art methods in textile testing. A more detailed discussion of the decision-making process of the DTSVM and its advantages over traditional methods could strengthen the manuscript.

3.The experimental setup and the data analysis are thoroughly described, providing a solid foundation for the claims made regarding the system accuracy and reliability. Including ten-fold cross-validation is commendable, as it enhances the credibility of the classification performance.

4.Additional information on the selection process for the fabrics tested would enhance the reader understanding of the system applicability to a wide range of textile materials.

5.The manuscript outlines the potential impact of the proposed system on improving the workflow and traceability of fabric tear performance testing processes. However, the discussion on practical implications is limited.

6.It would be valuable to expand on how this system can be integrated into existing textile testing laboratories and its compatibility with current industry standards. Additionally, outlining future research directions, especially concerning the system adaptability to new materials and testing conditions, would be beneficial.

7.While the manuscript is generally well-structured, some sections could benefit further clarification and refinement. For instance, the introduction of the DTSVM algorithm and its underlying principles could be made more accessible to readers not familiar with machine learning techniques.

8.The manuscript presents a significant contribution to textile testing, offering a robust method for the real-time monitoring and classifying of fabric tear performance states. With some revisions, particularly in enhancing the discussion on the system practical implications and future work, the manuscript could provide a valuable resource for researchers and practitioners in the textile industry.

The author’s answer:

Your recognition and encouragement give me unlimited confidence in my research, thank you very much for your suggestions, I will reply to your suggestions one by one:

1.Thank you for recognizing the innovative nature of my article.

2.For the multi-classification method of DTSVM, I also addedadded other classification methods (one-against-all (OAA) and one-against-one (OAO)) for experimental comparisons to conduct a more comprehensive evaluation of the model proposed in this paper, which is reflected in the chapter of "Experimental Results and Discussion". Upon comparison, the classification accuracy of DTSVM is much higher than other methods, which further validates the feasibility and accuracy of the method in this paper.

3.The ten-fold cross validation is an example of increasing the generalization of the method, and I thank you for highly endorsing the use of this validation method.

4.In this paper, 6 out of 6 fabrics with different elasticity were used in the experimental part and the accuracy of classification was more than 98.73%, which shows the applicability of the method of this paper.

5.In the section entitled "Conclusion", a new meaning has been added for practical significance: "It also provides new ideas and solutions for the application of intelligent perception technology in the field of traditional textile inspection".

6.In the "Outlook" section, it is explained in detail that the system and experimental model of this paper need to be updated with the standard update, and the anomalies and predictions in the experimental process will be investigated in the next article, so as to adapt to more experimental scenarios.

7.The principle of DTSVM has been added in "Decision Tree Support Vector Machine" , which is convenient for readers to understand.

8.The practical significance of the system and the discussion aspects of future work have been shown in the "Conclusion" section, and the detailed modifications have been highlighted in the revised version.

Thank you very much for your attention and time. Look forward to hearing from you.

---

## [Decision Letter · Decision Letter 1]

28 Mar 2024

Fabric tearing performance state perception and classification driven by multi-source data

PONE-D-24-02493R1

Dear Dr. Jiao,

We’re pleased to inform you that your manuscript has been judged scientifically suitable for publication and will be formally accepted for publication once it meets all outstanding technical requirements.

Kind regards,

Praveen Kumar Donta, Ph.D.

Academic Editor

PLOS ONE

Additional Editor Comments (optional):

Reviewers' comments:

Reviewer's Responses to Questions

**Comments to the Author**

1. If the authors have adequately addressed your comments raised in a previous round of review and you feel that this manuscript is now acceptable for publication, you may indicate that here to bypass the “Comments to the Author” section, enter your conflict of interest statement in the “Confidential to Editor” section, and submit your "Accept" recommendation.

Reviewer #2: All comments have been addressed

Reviewer #3: All comments have been addressed

2. Is the manuscript technically sound, and do the data support the conclusions?

Reviewer #2: Yes

Reviewer #3: Yes

3. Has the statistical analysis been performed appropriately and rigorously? 

Reviewer #2: Yes

Reviewer #3: Yes

4. Have the authors made all data underlying the findings in their manuscript fully available?

Reviewer #2: Yes

Reviewer #3: Yes

5. Is the manuscript presented in an intelligible fashion and written in standard English?

Reviewer #2: Yes

Reviewer #3: Yes

6. Review Comments to the Author

Reviewer #2: I have reviewed the entire manuscript once again, it looks good. Thank you for implementing the recommended changes.

Reviewer #3: I have no further comments. In my opinion authors address all changes as required by reviewers. I believe that paper can be accepted in actual state.

7. PLOS authors have the option to publish the peer review history of their article (what does this mean?). If published, this will include your full peer review and any attached files.

Reviewer #2: No

Reviewer #3: No

---

## [Editor Report · Acceptance letter]

3 Apr 2024

PONE-D-24-02493R1 

PLOS ONE

Dear Dr. Jiao, 

I'm pleased to inform you that your manuscript has been deemed suitable for publication in PLOS ONE. Congratulations! Your manuscript is now being handed over to our production team.

Kind regards, 

on behalf of

Dr. Praveen Kumar Donta 

Academic Editor

PLOS ONE